# Diagnostic Performance and Clinical Utility of Conventional PCR Assay in Early Diagnosis of COVID-19 Associated Rhino-Orbito-Cerebral Mucormycosis

**DOI:** 10.3390/jof8080844

**Published:** 2022-08-11

**Authors:** Samir Mohapatra, Manas Ranjan Barik, Suryasnata Rath, Savitri Sharma, Archisman Mohapatra, Sharmistha Behera, Souvagini Acharya, Dipti Ranjan Pattjoshi, Rajesh Kumar Padhi, Himansu Sekhar Behera

**Affiliations:** 1Department of Ophthalmic Plastic and Reconstructive Surgery Service, L V Prasad Eye Institute, Mithu Tulsi Chanrai Campus, Bhubaneswar 751024, Odisha, India; 2Ocular Microbiology Service, L V Prasad Eye Institute, Mithu Tulsi Chanrai Campus, Bhubaneswar 751024, Odisha, India; 3Jhaveri Microbiology Centre, L V Prasad Eye Institute, Kallam Anji Reddy Campus, Hyderabad 500034, Telangana, India; 4Department of Biostatistics, Generating Research Insights for Development (GRID) Council, Noida 201307, Delhi NCR, India; 5Department of Ophthalmology, VSSIMSAR, Burla 768017, Odisha, India; 6Department of Ear, Nose and Throat, VSSIMSAR, Burla 768017, Odisha, India; 7Department of Ear, Nose and Throat, SCB Medical College and Hospital, Cuttack 753007, Odisha, India; 8Department of Ear, Nose and Throat, Sparsh Hospitals Pvt Ltd., Bhubaneswar 751007, Odisha, India

**Keywords:** mucormycosis, nasal biopsy, endonasal swabs, KOH + CFW smear, PCR assay, mixed infection

## Abstract

Early diagnosis and treatment of rhino-orbital-cerebral mucormycosis (ROCM) are crucial. Potassium hydroxide with Calcofluorwhite (KOH + CFW) smears can demonstrate the fungal hyphae, but mixed infections caused by both mucorales and non-mucorales pose a diagnostic challenge. Polymerase chain reaction (PCR) can detect mixed infections and differentiate mucorales from non-mucorales. This study aimed to evaluate the utility of a single reaction PCR in the diagnosis of ROCM and the efficacy of nasal biopsy and endonasal swab in the detection of fungus. **Sixty-six** clinical samples were collected from 33 patients and were subjected to KOH + CFW smear, culture and PCR. PCR was performed using pan-fungal primers targeting the 28S large subunit rRNA gene, and the amplified products were further sequenced to identify the fungi. KOH + CFW smear, culture and PCR detected mucorales in 54.6%, 27.3% and 63.6% patients, respectively. PCR detected mixed infection in 51.5% patients compared to 9.1% by KOH + CFW smear. PCR detected fungus in 90% of nasal biopsies and 77.8% of endonasal swabs. *Rhizopus* spp. was the most common fungi identified in 43.2% of PCR-positive samples. PCR is effective in detecting mixed infection and in the diagnosis of ROCM. Nasal biopsies had better fungal detection rates than endonasal swabs.

## 1. Introduction

Rhino-orbital-cerebral mucormycosis (ROCM) is an angioinvasive disease with high morbidity and mortality. There was a sudden surge in the number of ROCM cases during the second wave of the COVID-19 pandemic in many countries, including India [1]. Diabetes mellitus, corticosteroid therapy, immunosuppression, hematological and solid organ malignancies, hematological stem cell and solid organ transplantation, and iron overload are the major risk factors [2]. India has the highest burden of COVID-19-associated mucormycosis (CAM) in the world, with an estimated prevalence of 140 cases per million population [1,3,4,5].

Mucormycosis is caused by fungus in the order mucorales, among which *Rhizopus* spp. are the predominant ones, followed by *Mucor* spp. and *Lichtheimia* spp. [6]. Among the non-mucorale fungi, *Aspergillus flavus* and *Candida albicans* are the predominant ones associated with rhino-orbital-cerebral infections. Mixed infections caused by both mucorales and non-mucorales and those caused by other opportunistic pathogenic fungi in immunocompromised patients cannot be ignored [7]. Early diagnosis and treatment with accurate antifungal therapy are crucial in the management of ROCM. A delay in diagnosis and treatment may increase the morbidity and mortality to 50–80% due to angioinvasiveness of the disease [8]. Definitive identification of the causative organism is also important as it may guide the choice of antifungal therapy. Mucorales are specifically susceptible to Amphotericin B, while non-mucorale fungi are more susceptible to the azole group of antifungal drugs [9]. The risk of serious adverse effects associated with these drugs should also be considered during treatment. Hence a diagnostic modality that can provide a definitive early diagnosis is desirable. Potassium hydroxide with Calcofluor white (KOH + CFW) stained smears demonstrates fungi well and can differentiate mucorales from non-mucorale fungi based on the hyphal characteristics [2]. This differentiation, however, becomes indecisive in the presence of broken fungal filaments, atypical fungal hyphae or the presence of mixed septate and aseptate hyphae in the clinical samples [10]. In such cases, the growth of fungus in culture media provides a definitive diagnosis and is considered the gold standard. However, it usually takes 2–7 days for isolation of the fungus in culture and may take longer for identification. Culture has a low diagnostic yield, which further decreases with the increase in the time interval between sample collection and inoculation. Molecular methods such as polymerase chain reaction (PCR) assay can accurately identify the fungal pathogen in clinical samples within a short time. However, using two different genus-specific primers and two separate PCR reactions to differentiate mucorale from non-mucorale fungi is time consuming [11].

The present study was conducted to evaluate the utility of a single reaction conventional PCR assay in identifying fungal pathogens in ROCM patients and simultaneously differentiating mucorales from non-mucorale fungi. In addition, this study compared the efficacy of nasal biopsy and endonasal swab for the isolation of fungi in ROCM patients. 

## 2. Methods

### 2.1. Study Design

This is an observational, non-comparative, cross-sectional study conducted between May and December 2021 at the L V Prasad Eye Institute, a tertiary eye care center in Odisha, Eastern India. Three tertiary care general hospitals within the State of Odisha collaborated on this study. The study included COVID-19 recovered patients who had warning signs and symptoms suggestive of possible ROCM infection as described earlier [2]. All patients had recovered from a recent COVID-19 infection (≤6 weeks), did not have any active COVID-19 symptoms and had tested negative by Rapid antigen test (RAT) or Real-time reverse transcription polymerase chain reaction (RT-PCR) at presentation. Patients with active COVID-19 symptoms, who tested positive by RAT or RT-PCR tests and who had received systemic (oral or intravenous) antifungal therapy in the preceding 6 weeks of presentation were excluded from this study. The study protocol followed the principles of the Declaration of Helsinki and was approved by the Institutional Ethics Committee (2021-102-BHR-41). Written consent in vernacular language was taken from all study patients for inclusion in the study, clinical examination and sample collection. The schematic for the study is presented as a flow chart (Figure 1).

### 2.2. Clinical Data of Patients and Staging

The patients presented either to the tertiary eye care (study center) or to the three collaborating general hospitals. At the presentation, a detailed clinical history was taken, and a comprehensive ocular examination was performed on all patients. The clinical features, computed tomography (CT) and/or magnetic resonance imaging (MRI) findings and the risk factors were noted. The extent and severity of the disease were assessed as per the ROC staging system described by Naik et al. [12]. Each component of the disease, i.e., rhino (Nasal cavity and sinuses), orbital (Eye and Orbit) and cerebral (cavernous sinus, meninges and intracerebral) was graded between 0 and 3, and the stage of the disease was described as R 0-3 O 0-3 C 0-3. The severity of the disease was described as mild, moderate and severe [12]. In order to facilitate data analysis, the authors in this study categorised mild disease as any R (0,1a, 1b) O (0,1) C (0), moderate disease as any R (2a, 2b) O (2a, 2b) C (0) and severe disease as any R (3a, 3b) O (2c, 3a, 3b) C (1, 2a, 2b, 3a, 3b) (Table 1).

### 2.3. Surgical Intervention and Sample Collection

At presentation, diagnostic nasal endoscopy was performed in all patients under topical anesthesia (TA) using 10% Lidocaine spray (10 mg/puff; Neon Laboratories, India), and endonasal swabs were collected from the affected side. Depending on the severity of the disease, debridement of the necrotic tissues in the nasal cavity, paranasal sinuses, ocular adnexa, orbit and/or brain was performed under general anesthesia (GA) within 24 to 48 h of presentation and tissue biopsy samples were collected. All the samples were transferred to the microbiology laboratory of the L V Prasad Eye Institute within 2 h for in-house samples and within 24 h of collection from the general hospitals. 

Sixty-six clinical samples (27 endonasal swabs, 30 nasal biopsies, 7 orbital biopsies, 1 brain and lid biopsy each) were collected from 33 patients included in this study. Endonasal swabs were also collected from the contralateral normal nasal cavity of 17 patients and labeled as control swabs.

### 2.4. Conventional Microbiology Examination

All clinical samples and control swabs were examined by KOH + CFW smear, culture and PCR assay. All tissue samples and endonasal swabs were inoculated on blood agar (BA) and Sabouraud’s dextrose agar (SDA) and incubated at 37 °C and 29 °C, respectively, until growth was seen. Fungus growth was identified by their morphological characteristics in lactophenol cotton blue (LPCB) mount.

### 2.5. Standardisation of PCR Assay

DNA was extracted from the clinical isolates of 6 culture-positive patients; (03 *Rhizopus* spp. and 1 each of *Asperigillus flavus*, *Asperigillus niger* and *Candida albicans*) using QIAamp DNA Mini Kit (QIAGEN, GmbH, Hilden, Germany) following the manufacturer’s instructions. Broad-range PCR assay was standardized to amplify a partial region of the 28S large subunit rRNA gene of mucorales and other non-mucorale fungi in a single reaction. The primers used in the PCR assay were a published set of pan-fungal primers (FP: 5′GTGAAATTGTTGAAAGGGAA3′ and RP: 5′GACTCCTTGGTCCGTGTT3′) with modified PCR parameters described by us earlier [13]. DNA extracted from the confirmed isolates (by DNA sequencing) of *Rhizopus oryzae* and *Aspergillus flavus* was used as a positive control for mucorales and non-mucorale filamentous fungi, respectively. Appropriate negative control was included. Amplified PCR products were subjected to 1.5% agarose gel electrophoresis and visualized under a gel documentation system (Biorad, Hercules, CA, USA).

### 2.6. Analytical Sensitivity and Specificity of Primers

The analytical sensitivity of the primers was estimated by PCR with different concentrations of *Rhizopus oryzae* and *Candida albicans* DNA. The analytical specificity of the primers was determined by PCR with DNA of other predominant ocular pathogens such as *Staphylococcus aureus* ATCC 25923, *Escherichia coli* ATCC 25922, *Varicella zoster virus*, *Herpes simplex virus 1* and *Cytomegalovirus*.

### 2.7. Nucleotide Sequencing and Homology Analysis

Nucleotide sequencing was performed for all PCR-positive clinical samples. Amplicon with a single band were directly sequenced (Eurofins, Applied Biosystems 3730xl DNA Analyzer; Agri Genome Labs, Kochi, India, ABI). Amplicon with 2 bands were excised individually, purified with a gel extraction kit (Qiagen, Hilden, Germany) and sequenced separately. The obtained nucleotide sequences were searched for homology analysis with the available sequences of the 28S rRNA gene of the GenBank using the NCBI BLAST (NCBI, Rockville Pike, Bethesda, MD, USA) computer program (http://www.ncbi.nlm.nih.gov/pubmed; accessed on 21 March 2022). Nucleotide sequences obtained were submitted to NCBI, and accession numbers were obtained.

### 2.8. Data Compilation and Statistical Analysis

Data related to demographics, clinico-radiological features, co-morbidities and risk factors, type of clinical samples, microbiology results, treatment and outcomes were collected. Data anonymity, confidentiality, privacy and restricted access to electronic medical records (EMR) were maintained at all steps. Sensitivity, specificity and predictive values of PCR assay were calculated with culture as a gold standard. The same was also performed for KOH + CFW smear results. The accuracy of PCR and smear as diagnostic tests was compared using a Receiver operating characteristics (ROC) curve. Series and parallel combinations of KOH + CFW smear and PCR were assessed for sensitivity and specificity. The efficacy of endonasal swab and nasal biopsy for isolation of fungi in ROCM patients were also analyzed using Fisher’s exact test, and *p* < 0.05 was considered to be significant. 

## 3. Results 

### 3.1. Demographic and Clinical Details

The demographic details and clinical manifestations of patients are highlighted in Appendix A. Of the 33 patients, 72.7% (*n* = 24) were males and 9 (27.3%) were females. All patients had unilateral involvement; the right eye was more commonly affected (*n* = 17; 51.5%). The mean age of the patients was 50.78 ± 11.05 years (Range: 20–72 years; Median: 50 years). The median duration of COVID symptoms was 15 days (Mean: 16.5 ± 10.83 days; Range 4–60 days). Sixteen out of thirty-three (48.5%) patients were admitted to the hospital for COVID-19 treatment before they attended our institute for ophthalmological evaluation with clinical features suggestive of ROCM. The mean duration of hospitalization was 11.56 ± 3.92 days (Median: 12 days; Range 5–21 days). The median time interval between diagnosis of COVID-19 and onset of symptoms of ROCM was 14 days (Mean: 15.42 ± 8.36 days; Range: 3–40 days). The median time interval between onset of symptoms of ROCM to presentation was 10 days (Mean: 13.27 ± 10.21 days; Range: 1–42 days). Sixteen of thirty-three (48.5%) patients presented to the tertiary eye care center, and 17 (51.5%) patients reported to the collaborating general hospitals with symptoms of ROCM. Of the 33 patients, 17 (51.5%) had deranged blood sugars, 15 (45.5%) had received systemic corticosteroids (Mean duration: 8.8 ± 6.08 days; Median: 7 days; Range: 4–30 days), 12 (36.4%) had received non-invasive supplemental oxygen therapy (Mean duration: 7.58 ± 2.63 days; Median: 7 days; Range: 4–14 days) and four patients (12.1%) needed invasive ventilatory support for a mean duration of 7 days during COVID-19 treatment.

Of the 33 patients, 45.4% (*n* = 15) had mild, 18.2% (*n* = 6) had moderate and 36.4% (*n* = 12) had severe ROCM disease, the representative photographs are shown in Figure 2.

### 3.2. Diagnostic Performance of KOH + CFW Smear

Identification of non-septate or pauci septate, irregular, ribbon-like, broad hyphae in clinical samples in KOH + CFW smear suggested mucorales (Figure 3a). Thin and regularly septate hyphae with parallel walls were suggestive of non-mucorale filamentous fungi (Figure 3b). The presence of both aseptate and septate fungal filaments and/or budding yeasts in the same sample was suggestive of mixed infection (Figure 3c). Of the 33 patients, KOH + CFW smear identified mucorales in 45.4% (n = 15), non-mucorale fungi (filamentous fungi and budding yeasts) in 12.1% (n = 4) and mixed infection in 9.1% (n = 3) patients. In clinical samples, mucorales were identified in 15 of 30 (50%) nasal biopsies and 2 of 27 (7.4%) endonasal swabs. The KOH+ CFW smear results for all the clinical samples and control swabs are highlighted in Table 2. The turnaround time to provide the KOH + CFW smear report was approximately 1 h. 

### 3.3. Diagnostic Performance of Culture

White cottony colony turning into gray, black or brown colonies in culture media was suggestive of mucorale group of fungus. Microscopic examination under lactophenol cotton blue (LPCB) mount showing broad, aseptate hyphae with ribbon-like folds and brown or black sporangia were confirmatory of mucorales (Figure 3d). The growth of suede or velvety white colonies was suggestive of a non-mucorale group of fungus (Figure 3e), while the presence of both mucorale and non-mucorale colonies in the same sample was suggestive of mixed infection (Figure 3f). Culture showed growth of fungus in 16 of 33 (48.5%) patients of which mucorale was detected in 37.5% (n = 6), non-mucorale fungi in 43.8% (n = 7) and mixed infection in 18.8 % (n = 3) culture-positive patients. The culture showed growth of fungus in 23 of 66 (34.9%) clinical samples. On LPCB mount, mucorale fungi was identified in 39.1% (n = 9), non-mucorale fungi in 43.5% (n = 10) and mixed infection in 8.7% (n = 2) of the culture positive samples. Fungal isolates could not be identified on the LPCB mount in 8.7% (n = 2) samples. *Rhizopus* spp. was the commonest mucorale (43.5%; n = 10), and *Aspergillus* spp. was the predominant non-mucorale (21.7%; n = 5) fungi identified in the 23 culture-positive samples. The culture results for all the clinical samples and control swabs are highlighted in Table 2. The turnaround time to establish the culture diagnosis was approximately 48–144 h.

### 3.4. Diagnostic Performance of PCR Assay 

In PCR assay, an intense band of ~340 bp amplicon size was obtained from genomic DNA of three *Rhizopus* spp., and a relatively smaller ~259 bp amplicon size was obtained from genomic DNA of the three non-mucorale fungi (*Asperigillus flavus*, *Asperigillus niger* and *Candida albicans*) (Figure 4a). The limit of detection of the PCR assay was 6.25 ng and 5.0 ng, respectively, for *Rhizopus oryzae* and *Candidida albicans*. The analytical specificity of the PCR assay is shown in Figure 4b. PCR assay was positive for fungal pathogens in 31 of the 33 (93.9%) patients.

Mucorales were identified in 9.7% (n = 3), non-muocrale fungi in 35.5% (n = 11) and mixed infection in 54.8% (n = 17) of the PCR positive patients. Fifty-seven of sixty-six (86.4%) clinical samples were PCR positive for fungal pathogens. The diagnostic performance of smear, culture and PCR assay are highlighted in Table 2.

### 3.5. Nucleotide Sequencing 

Nucleotide sequencing was performed for all the 57 PCR positive samples; of which identification of the organism up to the species level was performed in 37 (64.9%) samples; of which *Rhizopus* spp. were identified in 16 of 37 (43.2%), *Aspergillus* spp. in two (5.4%), opportunistic fungal pathogens in 16 (43.2%) and mixed infection in three (8.1%) samples. In the rest, 20 sample species could not be identified. The details of nucleotide sequencing are summarised in Appendix A.

### 3.6. Efficacy of Nasal Biopsy and Endonasal Swab in Detection of Fungal Pathogens 

Of the 33 patients, endonasal swabs and nasal biopsies were collected from 27 (81.8%) and 30 (90.9%) patients, respectively. In 25 patients, both the samples were collected, and their efficacy in the detection of fungal isolates was compared (Table 3). Detection of fungal isolates was significantly better in nasal biopsies than in endonasal swabs in KOH + CFW smear (*p* < 0.001), but there was no significant difference in the isolation rates between the two in PCR assay (*p* = 0.247).

### 3.7. Comparison of the Diagnostic Performance of Various Tests 

The sensitivity (Sn), specificity (Sp) and predictive values of KOH + CFW smear and PCR assay in comparison to culture are shown in Table 4. KOH + CFW smear and PCR assay had a kappa value of 0.05 (*p* = 0.676), suggesting a lack of agreement between the two tests. The area under the ROC curve for KOH + CFW and PCR assay was found to be 0.77 and 0.69, respectively (Figure 5). It is well-known that when two tests are applied in series, i.e., the tests are performed sequentially; the samples which test positive in both the tests are considered disease positive, and those who test negative in both the tests are considered disease negative. This increases the Sp and the positive predictive value (PPV) and decreases the Sn than each test applied in isolation. When applied in parallel, i.e., the tests are performed simultaneously, and each sample is subjected to both tests irrespective of the results in either, then the Sn of the combination is higher, and the Sp is lower than any of the individual tests applied in isolation.

In our study for mild cases, when the tests were applied in series, the Sn was 100% and Sp 83.2%, and when applied in parallel, Sn was 100%, but Sp was equal to 29.8%. For moderate-to-severe disease, when the tests are applied in series, the Sn was 65.3% and Sp 88.7%, and in parallel, Sn was 96.8% and Sp was 39.3%.

### 3.8. Treatment Outcomes 

Surgical debridement of the necrotic tissues in the nasal cavity, orbit, adnexal tissue and/or brain was performed in 30 of 33 (90.9%) patients under GA. Two patients were critically ill, and GA was contraindicated; hence debridement could not be performed. One patient had mild disease without any necrotic focus hence debridement was not advised. Patients with infection with mucorales and those with mixed infections were treated with intravenous liposomal amphotericin B or amphotericin B lipid complex (5mg/kg/day starting dose increased to 10 mg/kg/day for 1 week) as per the recommended guidelines [2,14]. Depending on the clinical response at 1 week, further treatment was continued with either intravenous amphotericin B (5 mg/kg/day) or oral posaconazole (tablets 300 mg twice daily) till symptoms resolved. The patients with infection with only non-mucorale filamentous fungi were treated with intravenous voriconazole (6 mg/kg/day on day 1 followed by 4 mg/kg/12 h for 1 week). Depending on the clinical response at 1 week, further treatment was continued with oral voriconazole (tablets 200 mg twice daily) till symptoms resolved. Liver and kidney function tests were performed at the initiation of therapy and at biweekly intervals during the treatment period. Patients with a disease-free state at 3 months post-treatment follow-up were considered a treatment success. Progression of the disease and mortality was considered a treatment failure. Thirty of the thirty-three (90.9%) patients in this study were successfully treated and recovered from the disease; three patients died due to cerebral extension.

## 4. Discussion

Rhino-orbital-cerebral-mucormycosis is the most common presentation of all mucormycosis cases, followed by cutaneous, pulmonary and disseminated mucormycosis [15,16,17,18,19]. The median age of patients with CAM was reported as between 44.5 and 57 years, with a male predilection between 60 and 79% in various studies [16,17,18,19,20]. In our study, the median age of patients was 50 years, with 72% male predominance. The clinical manifestations of our patients were comparable to large series of CAM reported from India by Aurora et al. [20] and Sen et al. [21]. Poorly controlled diabetes mellitus is considered the most common risk factor COVID-19 associated ROCM infection. Twenty-two of thirty-three (66.7%) patients in our study had uncontrolled diabetes as the major risk factor as in other studies; however, the association of diabetes with the severity of ROCM was not significant (*p* = 0.24) [1,3,4,8].

KOH + CFW smear provides an early, presumptive diagnosis of ROCM based on the hyphal characteristics, but samples with smaller or fragmented fungal filaments pose a diagnostic challenge [2,14,22,23]. Culture depends on viable fungal filaments and has an isolation rate of 60–68% for mucorales [10,21,22,23]. In our study, mucorales were identified in 27.27% of patients in culture, in 54.54% of patients in KOH + CFW smear and in 63.64% of patients in PCR assay. The low isolation rate of mucorales in culture may be due to the damage of long, delicate and fragile hyphae of mucorales during sampling [10,21,22,23].

In our study, KOH + CFW smear was negative for fungus in 11 of 33 (33.3%) patients. PCR assay identified the fungal pathogens in all these KOH + CFW negative patients; 6 of 11 (54.5%) patients had non-mucorale fungal infection, and 5 (45.5%) had mixed infection. This advantage of PCR was perceived in one of our patients (SL No 17, Appendix A) who had severe ROCM and cerebral extension (Figure 2g). A brain biopsy from this patient showed no organism on the KOH + CFW smear. However, on PCR assay, mucorales were identified; the patient was treated aggressively and recovered from the disease.

In mixed infection cases, identification of the fungus becomes difficult both by KOH + CFW smear and culture. In KOH + CFW smear, short and slender hyphae of non-mucorales remain underneath the broad and long hyphae of mucorales making identification of non-mucorales difficult [22,23]. In culture, mucorales grow faster and mask the growth of non-mucorales [22,23]. In our study, PCR assay identified mixed infection in 17 of 33 (51.5%) patients; of which six (35.3%) were KOH + CFW negative, and seven (41.2%) were culture negative. This large proportion of patients would have missed the diagnosis based only on smear or culture reports.

In this study, KOH + CFW smear and PCR were comparable in terms of their respective Sn and Sp, which were also comparable with other studies [23,24]. Application of the tests in series (only those samples which test positive in KOH + CFW smear are subjected to PCR assay) led to high Sp (>83%) for ROCM disease across severity spectrum with high Sn (100%) for mild cases and Sn (>65%) for severe cases. Interpretation of these results may help earlier detection of even milder diseases and initiate appropriate treatment. In addition, this may guide the clinicians to avoid unnecessary use of a nephrotoxic drug such as Amphotericin B. Application of the tests in parallel (KOH + CFW and PCR performed simultaneously) increases the Sn to 100%. This increases the chances of detection of fungal infection and ensures that the diagnosis is not missed in the clinical samples. In addition to the PCR’s ability to differentiate mucorales from non-mucorales, this can guide clinicians in initiating appropriate antifungal therapy and decrease morbidity.

Our study found that fungal pathogens were detected in a higher proportion of nasal biopsies (66.7% in smear vs. 90% in PCR) compared to endonasal swabs (14.8% in smear vs. 77.8% in PCR). For diagnostic accuracy, both nasal biopsy and endonasal swabs should be collected wherever feasible. However, in critically ill patients in whom GA is contraindicated, such as in two of our patients (SL No 27 and 28, Appendix A), only endonasal swabs can be collected and subjected to tests for fungus identification. It is relatively simpler, faster and can be performed under TA even at the bedside of patients. It can also be collected from patients at rural health centers and transferred to higher centers for definitive diagnosis. This may help initiation of therapy, avoiding the delay in referral of patients.

The PCR technique highlighted in this study is simpler, needs minimal training and is less costly than real-time PCR. It utilizes a single reaction to differentiate mucorale from non-mucorale fungi consuming less time. This technique can be utilized in resource constraint health care facilities for making an early diagnosis of ROCM infection. This study has a few limitations. The small sample size may be due to the low incidence of COVID-19 ROCM infection in our geographical area. Due to logistic issues prevailing during the second wave of the COVID-19 pandemic, there was no uniformity in sample collection, which otherwise would have allowed a head-on comparison between KOH + CFW smear, culture and PCR assay results.

## 5. Conclusions

KOH + CFW smear and PCR assay are effective in the early diagnosis of ROCM. PCR assay has added advantage of detecting fungal infections in KOH-negative as well as culture-negative cases; it is also useful in detecting mixed fungal infections. Though nasal biopsies have a better chance of detecting fungi in KOH + CFW smear and PCR assay, endoansal swabs are particularly helpful in critically ill patients. When considering the diagnostic performance, this study recommends the use of PCR assay along with KOH + CFW smear to establish an early diagnosis of ROCM.

## Figures and Tables

**Figure 1 jof-08-00844-f001:**
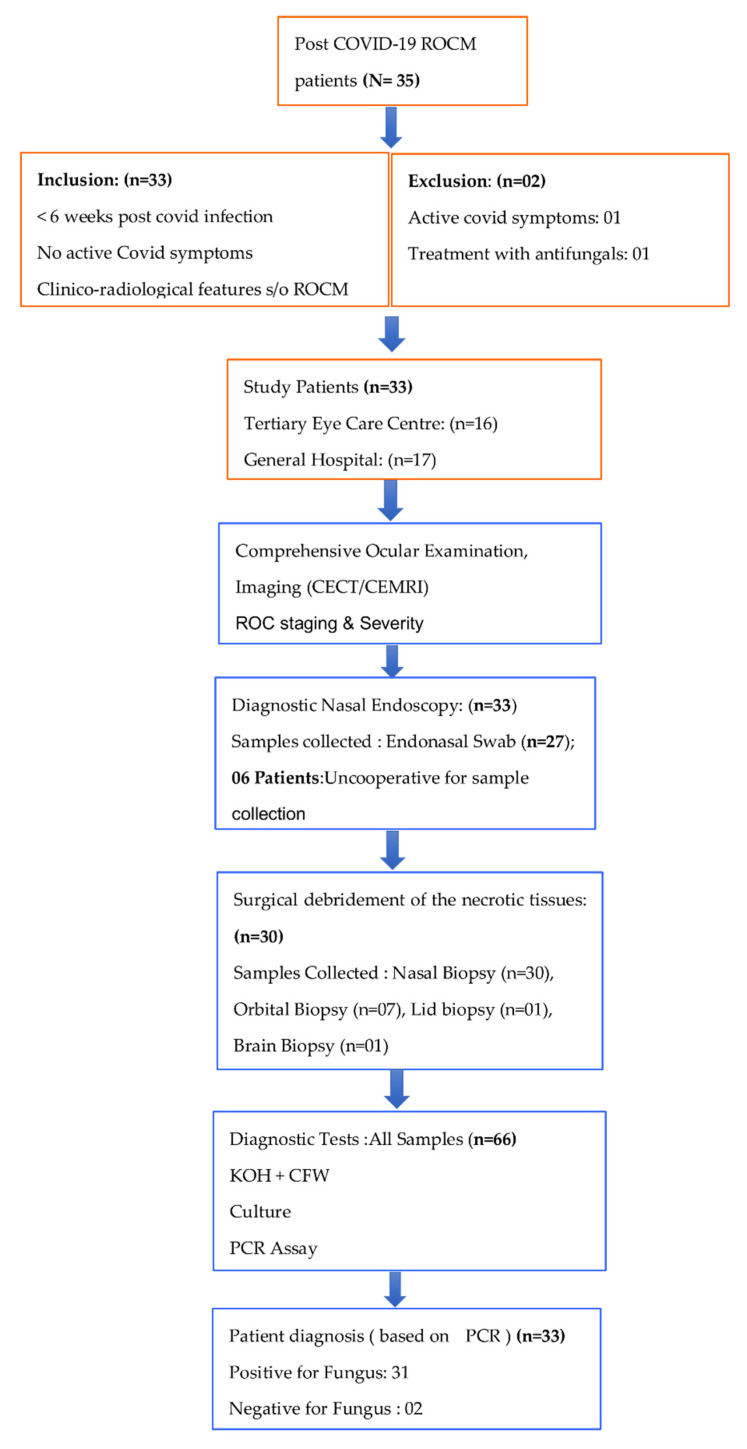
Flow chart depicting the schematic of the study.

**Figure 2 jof-08-00844-f002:**
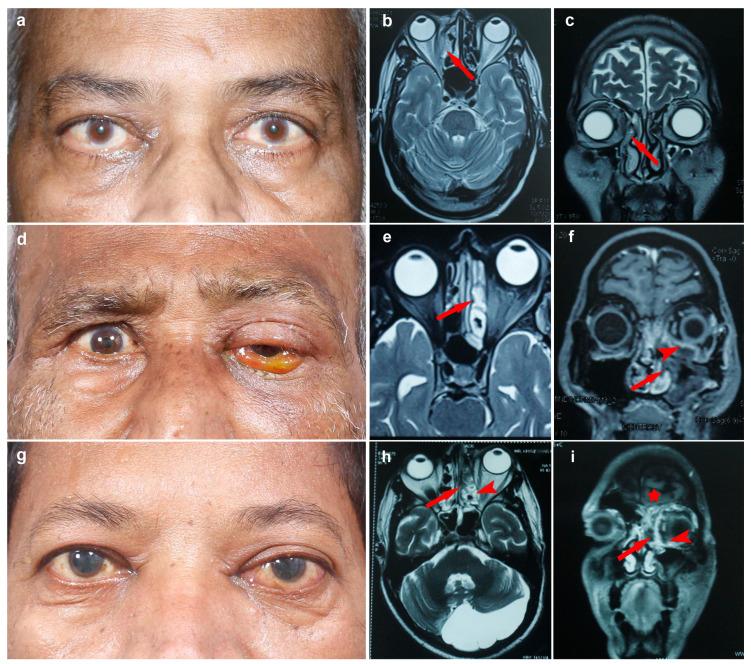
**Mild Disease** (**a**–**c**): 64 years male with lid edema, mild proptosis, BCVA 20/20 in right eye (**a**). T2W MRI images (**b**,**c**) show thickening of ethmoid sinus (red arrow) without orbital involvement. **Moderate Disease** (**d**–**f**): 67 years male patient with redness, lid edema, proptosis, BCVA 20/40 in the left eye (**d**). T2W and T1W MRI (**e**,**f**) images showing ethmoid and maxillary sinusitis (red arrow) with focal inferomedial orbital involvement (red arrowhead). **Severe Disease** (**g**–**i**): 57 years male with severe headache with pain, redness, proptosis, restricted ocular movements, BCVA PL+/PR inaccurate in left eye (**g**). T2W and T1W MRI (**h**,**i**) images showing pansinusitis (red arrow) with diffuse orbital involvement (red arrowhead) and focal brain involvement (red asterix).

**Figure 3 jof-08-00844-f003:**
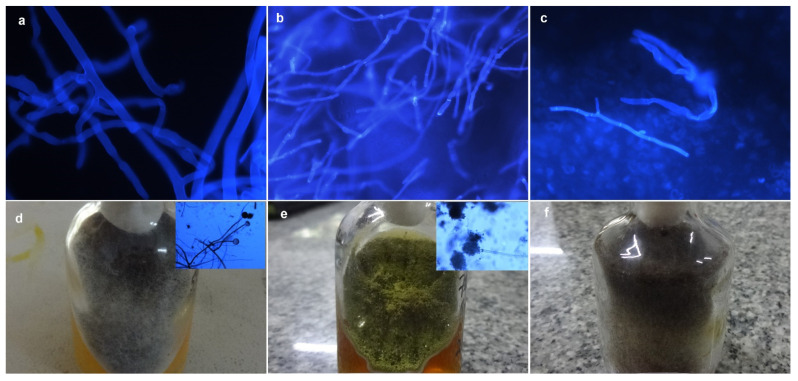
(**a**): KOH + CFW mount showing broad, aseptate fungal hyphae; (**b**): KOH + CFW mount showing narrow and septate fungal hyphae. (**c**): KOH + CFW mount showing both broad, aseptate and narrow, septate fungal hyphae in one sample. (**d**): Black, cottony fungus grown on SDA medium and Lactophenol cotton blue mount showing sporangia and rhizoides. (**e**): Yellowish green Fungus grown on SDA medium and Lactophenol cotton blue mount showing rounded onidial heads. (**f**): Both black, cottony fungus and yellow-green fungus in the same SDA medium.

**Figure 4 jof-08-00844-f004:**
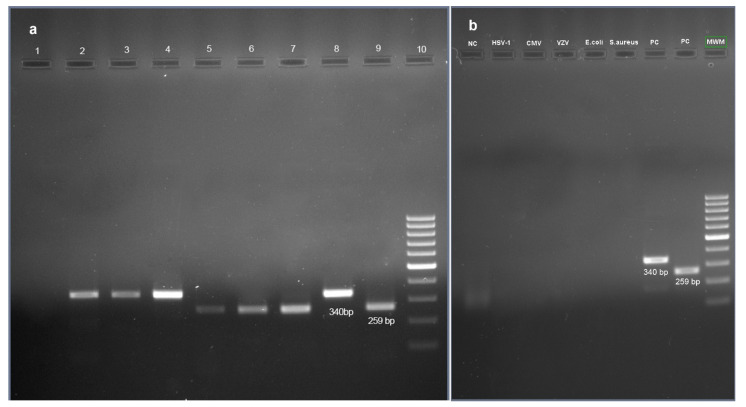
(**a**): Gel image showing a more intense band of 340 bp amplicon size of mucorales and less intense band of 259 bp amplicon size of the non-mucorales; (**b**): Gel image showing the specificity of PCR assay.

**Figure 5 jof-08-00844-f005:**
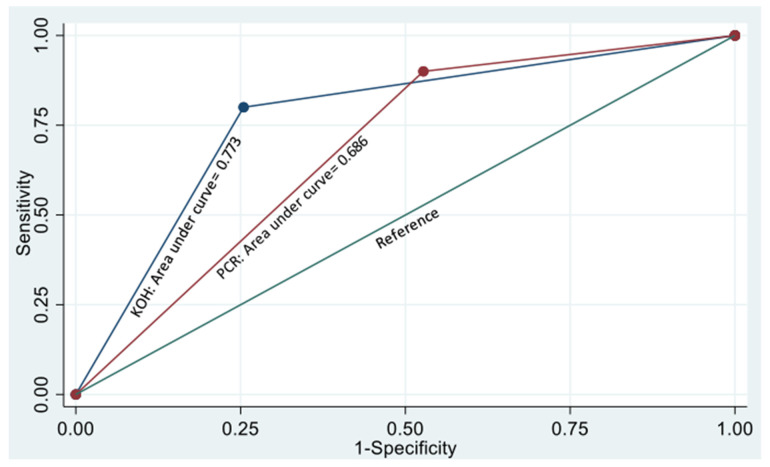
Receiver Operating Characteristic (ROC) Curve. KOH + CFW: ROC area: 0.7727. PCR: ROC area: 0.686.

**Table 1 jof-08-00844-t001:** ROCM Disease Severity Classification based on the ROC Staging of Naik et al. [12].

Disease Severity	ROC Gradng	Grade	Clinical + Nasal Endoscopy + Imaging Findings	No of Patients (*n* = 33) (%)
Mild	R (0, 1a, 1b)	0	Endoscopy normal, Imaging Normal	15 (45.45%)
		1a	Only Nasal mucosa involved, Sinuses clear	
		1b	Mucosal thickening of One or two sinuses	
	O (0, 1)	0	No orbital involvement, Vision Normal	
		1	Mild proptosis/movement restriction, One or Two EOMs enlarged, Vision normal	
	C (0)	0	No neurological signs, Cavernous sinus normal	
Moderate	R (2a, 2b)	2a	One or two sinuses hazy	06 (18.18%)
		2b	Unilateral pansinusitis with mucosal thickening	
	O (2a, 2b)	2a	Moderate proptosis/movement restriction medial/focal extra-conal orbital involvement with contrast enhancement	
		2b	Moderate proptosis/movement restriction medial/focal extra-conal orbital involvement with no contrast enhancement	
	C (0)	0	No neurological signs, Cavernous sinus normal	
Severe	R (3a, 3b)	3a	Unilateral Sinusitis, completely hazy sinuses	12 (36.36%)
		3b	Bilateral Sinusitis, complete or incompletely hazy	
	O (2c, 3a, 3b)	2c	Localised subperiosteal abscess	
		3a	Severe proptosis/complete ophthalmoplegia, No vision with diffuse orbital involvement	
		3b	Severe proptosis/complete ophthalmoplegia, No vision with diffuse orbital involvement with SOV thrombosis	
	C (1, 2, 3)	1	Focal Cavernous sinus involvement	
		2	Focal Cavernous sinus involvement with cavernous sinus thrombosis	
		3	Unifocal or multifocal CNS disease	

(Abr: ROC: Rhino-Orbito-Cerebral; SOV: Superior Ophthalmic Vein: EOMs: Extraocular Muscles, CNS: Central Nervous System).

**Table 2 jof-08-00844-t002:** Diagnostic performance of various tests in detecting fungal pathogens in clinical samples of ROCM patients.

		KOH + CFW			Culture			PCR Assay		
		M (+)	M + F (+)	F (+)	M (+)	M + F (+)	F (+)	M (+)	M + F (+)	F (+)
Sample Based Diagnosis (N = 83)	Nasal biopsy (n = 30)	15 (50.0%)	02 (6.7%)	03 (10%)	07 (23.3%)	01 (3.3%)	06 (20.0%)	03 (10.0%)	16 (53.3%)	08 (26.7%)
	Endonasal swabs (n = 27)	02 (7.4%)	01 (3.7%)	01 (3.7%)	0 (0%)	0 (0%)	03 (11.1%)	03 (11.1%)	11 (40.7%)	07 (25.9%)
	Orbit Biopsy (n = 7)	02 (28.6%)	0 (0%)	0 (0%)	02 (28.6%)	0 (0%)	03 (42.9%)	02 (28.5%)	01 (14.3%)	04 (57.1%)
	Brain Biopsy (n = 1)	0 (0%)	0 (0%)	0 (0%)	0 (0%)	0 (0%)	0 (0%)	01 (100%)	0 (0%)	0 (0%)
	Lid Biopsy (n = 1)	0 (0%)	01 (100%)	0 (0%)	0 (0%)	01 (100%)	0 (0%)	0 (0%)	01 (100%)	0 (0%)
	Control swabs (n= 17)	0	0	0	0	0	0	0	0	0
Patient Based Diagnosis (N = 33)		15 (45.5%)	03 (9.1%)	04 (12.1%)	06 (18.2%)	03 (9.1%)	07 (21.2%)	03 (9.1 %)	17 (51.5%)	11 (33.3%)

(Abr: M: Mucorales Group of Fungi, M + F: Mixed Infection with Mucorale and Non-Mucorale group of fungi, F: Non-Mucorales group of fungi).

**Table 3 jof-08-00844-t003:** Comparision of the efficacy of Endonasal swab and Nasal Biopsy in detecting fungal isolates in ROCM patients.

	Sample (No of Patients)	KOH + CFW Positive (%)	KOH + CFW Negative	*p* Value	PCR Positive	PCR Negative	*p* Value
Overall efficacy	Nasal Biopsy (n = 30)	20 (66.7%)	10 (33.3%)	NA	27 (90%)	03 (10%)	NA
	Endonasal Swab (n = 27)	04 (14.8%)	23 (85.2%)	NA	21 (77.8%)	06 (22.2%)	NA
Comparison of efficacy in select group of patients †	Nasal Biopsy (n = 25)	17 (68.0%)	08 (32.0%)	*p* < 0.001	23 (92.0%)	02 (8.0%)	*p* = 0.247
	Endonasal Swab (n = 25)	03 (12.0%)	22 (88.0%)		19 (76.0%)	06 (24.0%)	

(†: Group of patients in whom both Nasal Biopsy and Endonasal swabs were collected; n = 25).

**Table 4 jof-08-00844-t004:** Comparision of diagnostic performance of KOH + CFW smear and PCR assay (Sensitivity/Specificity).

Disease Severity		KOH + CFW Smear (95% CI)	PCR Assay (95% CI)	Tests in Series	Tests in Parallel
Mild	Sn	100.0%	100%	100%	100%
	Sp	71.4% (55.3–87.6%)	42.9% (25.2–60.6%)	83.2%	29.8%
	PPV	20% (5.7–34.3%)	11.1% (−0.13–22.4%)	-	-
	NPV	100.0%	100.0%	-	-
Moderate to severe	Sn	75% (60.9–89.1%)	87.5% (76.7–98.3%)	65.3%	100%
	Sp	78.6% (65.2–92.0%)	50.0% (33.7–66.3%)	89%	39.0%
	PPV	50.0% (33.7–66.3%)	33.3% (17.9–48.7%)	-	-
	NPV	91.7% (82.6–100%)	93.3% (85.2–101.5%)	-	-

(Abr: Sn: Sensitivity, Sp: Specificity, PPV: Positive Predictive Value, NPV: Negative Predictive Value, CI: Confidence Interval).

## Data Availability

The data supporting the results are available and can be obtained upon request by email to the corresponding author.

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
