# Peer review of "Diagnostic Performance and Clinical Utility of Conventional PCR Assay in Early Diagnosis of COVID-19 Associated Rhino-Orbito-Cerebral Mucormycosis"

_jof, 2022, doi:10.3390/jof8080844_

Round 1
Reviewer 1 Report
While diagnosis of mucormycosis is difficult, and authors suggest a PCR technique followed by nucleotide sequencing, I feel it is not ready for publication. I think the work is not presented in a manner with sufficient scientific merit.
There are many instances where the authors appear to have been careless in presenting the their own work. For instance, in the title they write: Diagnostic performance and clinical utility of conventional PCR assay in early diagnosis of COVID-19 associated rhino-orbito-cerebral mucormycosis. But in the methodology, the only included in the study, non-COVID-19 patients, or post-COVID-19 patients, and they did not explain why they chose that populations of patients and discarded the COVID-19 positive and active population, bearing in mind the high burden of CAM in India. However, I think that the usefulness of this study, should be in diagnosis Mucormycosis, independently of the COVID-19 status. Then, it is also no clear, how they differentiate colonization from current infection. The refered to mixed infections with Candida and Aspergillus, but it is possible, that this fungi were colinizating. So I think that authors should strenght their findings. A reference that showed that A. flavus and C. albicans are non-mucorale fungi freqeuntly associated with ROCM should be added. Reference 7 is not right in the context cited, because it referes only to COVID-19 patients. Similarly in line 42, where is not clear if authors are referring to 140 cases per million population on general or to the covid-19 population in India.
Other data about selected patients is also missing: are diabetic? which were they risk factors?
In line 67 they write: Molecular methods like polymerase chain reaction (PCR) assay can accurately identify the fungal pathogen in clinical samples within a short time. But using two different genus specific primers and two separate PCR reactions to differentiate mucorale from non-mucorale fungi is time consuming [11]. Ok, but the PCR method they also described, is also time consuming, because you should sequence the amplicons. And this is not a minor issue, since many laboratories from middle and low income countries have not available a DNA sequencer, and they should send the amplicons to another laboratory, so, maybe results are in one week. So, I think authors should make an effort to reinforce their results.
Author Response
Dear Sir
I am sending you the revised manuscript as an attachment also for your reference
Response to Reviewer 1 Comments:
Point 1: There are many instances where the authors appear to have been careless in presenting their own work. For instance, in the title they write: Diagnostic performance and clinical utility of conventional PCR assay in early diagnosis of COVID-19 associated rhino-orbito-cerebral mucormycosis. But in the methodology, the only included in the study, non-COVID-19 patients, or post-COVID-19 patients, and they did not explain why they chose that populations of patients and discarded the COVID-19 positive and active population, bearing in mind the high burden of CAM in India. However, I think that the usefulness of this study, should be in diagnosis Mucormycosis, independently of the COVID-19 status.
Response 1: Thanks a lot for your sincere query. We had thoughtfully designed the study methodology to include patients who developed symptoms of Rhino- orbital – cerebral mucormycosis (ROCM) following COVID-19 infection. This study inlcuded only COVID-19 recovered patients who were Rapid Antigen Test (RAT) or Reverse-Transcriptase PCR (RTPCR) negative at the time of initial presentation with features of ROCM. To justify we provide the following explanation:
- Our institute is a tertiary eye care centre located in the state of Odsha in India. There were several regulations and directives issued by the government of Odisha and other statutory bodies regarding managemnt of COVID-19. On such directive mentioned that all SARS COVID-19 positive patients who had active COVID symptoms shall be admittted initially to COVID Dedicated Hospitals (CDH) or COVID Dedicated Units (CDU) in a multispeciality hospital. The CDH or CDU were completely isolated from rest of the patient care areas and well equipped with isolation wards, intensive care units (ICU), ventilators and oxygen delivery systems. Active COVID-19 patients were initially treated for COVID symptoms and those who developed symptoms of ROCM during treatment were evaluated at the CDH. Patients who had recovered from COVID-19 infection (RAT or RT-PCR negative), were subsequently referred to our institute for ophthalmic manifestations. This small group of patients who presented to our institute with clinical features of ROCM were included in this study. They had already recovered from COVID-19 infection. Our institute did not have any COVID isolation ward or ICU facility for management of active COVID cases. The median time lag between COVID-19 diagnosis and onset of symptoms of Mucormycosis in our patients was 14 days
The same directives were also applicable to our collaborating multispeciality hospitals. Those hospitals have ICU and Oygen delivery systems and actively participated in the care of ROCM patients but were not designated as CDH or CDU by the government of Odisha .
- We also received several calls from CDH or CDU around our institute to evalaute the active COVID-19 patients who had developed features of ROCM. Though we actively participated in the clinical care of such patients, we couldnot collect any endonasal swabs and bring to the microbiology laboratory in our institute. The reason being our laboratory is a Biological safety Level (BSL) -2 laboratory. To handle infectious agents or toxins that may be transmitted through the air and cause lethal infections like SARS-COVID 19, BSL-3 laboratory is needed as per the Centre for Disease control and prevention biosafety level guidelines. So we collected samples only from COVID-19 recovered patients who presented to our tertiary eye care centre and to the two collaborating general hospitals.
Point 2: Then, it is also no clear, how they differentiate colonization from current infection. The refered to mixed infections with Candida and Aspergillus, but it is possible, that this fungi were colinizating. So I think that authors should strenght their findings. A reference that showed that A. flavus and C. albicans are non-mucorale fungi freqeuntly associated with ROCM should be added. Reference 7 is not right in the context cited, because it referes only to COVID-19 patients.
Response 2: Thanks a lot for your sincere query. The explanation is mentioned below:
- Rhino-orbital-cerebral mucormycosis (ROCM) are more common in immunocompromised patients and is caused by the fungus of the order mucorales. COVID-19 infection may induce significant and persistent lymphopenia which may increase the risk of opportunistic fungal infections with Mucor, Candida and Aspergillus. Mixed infection can caused by these three fungi in a suspected case of ROCM. In 2019 there was a published paper from India which clearly mentioned the co-existance of Mucor, Aspergillus and Candida in oculo-rhino-cerebral mycosis infection. ( Reference No 7). We have added the reference at serial No 7 to strengthen our findings. We have removed the existing reference at serial No 7 as suggested.
Added Reference No 7: Pandey D, Agarwal M, Chadha S, Aggarwal D. Mixed opportunistic infection with Mucor, Aspergillus and Candida in oculo-rhino-cerebral mycosis: An uncommon case. J Acad Clin Microbiol 2019;21:47-9
Removed Reference No 7: Song G, Liang G, Liu W. Fungal Co-infections Associated with Global COVID-19 Pandemic: A Clinical and Diagnostic Perspective from China. Mycopathologia. 2020 Aug;185(4):599-606.
Point 3: Similarly in line 42, where is not clear if authors are referring to 140 cases per million population on general or to the covid-19 population in India.
Response 3: Thanks for your query. The prevalence of Mucormycosis varies from 0.01 to 0.2 per 100 000 population in Europe and the United States of America and is much higher in India (14 per 100 000 population). Thus we have referred to 140 cases of Mucormycosis per million population on general and not to COVID-19 population.
Point 4: Other data about selected patients is also missing: are diabetic? which were they risk factors?
Response: This data has been mentioned as “Of 33 patients, 17 (51.5%) had deranged blood sugars, 15 (45.5%) had received systemic corticosteroids (Mean duration: 8.8 ± 6.08 days; Median: 7 days; Range : 4-30 days), 12 (36.4%) had received non-invasive supplemental oxygen therapy (Mean duration : 7.58 ± 2.63 days; Median: 7 days; Range: 4-14 days) and four patients(12.1%) needed invasive ventilatory support for a mean duration of 7 days during COVID-19 treatment.” In the result section. It is highlighted in yellow.
Point 5: In line 67 they write: Molecular methods like polymerase chain reaction (PCR) assay can accurately identify the fungal pathogen in clinical samples within a short time. But using two different genus specific primers and two separate PCR reactions to differentiate mucorale from non-mucorale fungi is time consuming [11]. Ok, but the PCR method they also described, is also time consuming, because you should sequence the amplicons. And this is not a minor issue, since many laboratories from middle and low income countries have not available a DNA sequencer, and they should send the amplicons to another laboratory, so, maybe results are in one week. So, I think authors should make an effort to reinforce their results.
Response 5: Nucleotide sequencing help in species level identification, which was tried in each sample in our study to verify the result. As stated in manuscript a band size of ~340 bp signifies the presence of mucoral group of fungus while an amplicon size of ~259bp signifies the presence of non-mucorale fungi in samples. The sequencing results preformed from ~340 bp amplicon size was proved to be of mucorale group of fungus with NCBI-BLAST in each sample. Similarly, sequencing results preformed from ~259 bp amplicon size was proved to be mucorale group of fungus with NCBI-BLAST in each sample. All obtained sequences were submitted to NCBI with accession numbers(mentined in manuscript). In few samples sequencing failed to to identify the species due to low intensity of bands.
Hence, it was proved that, primers used and PCR condition was able to differentiate between mucoral and non-mucoral group of fungus in a single PCR. PCR reaction followed with agarose gel electrophoresis will take around 3 hours for result interpretation. It is true that, since many laboratories from middle and low income countries do not have DNA sequencer, and they might send the amplicons to another laboratory. So, there may be a delay of 1 week in obtaining the sequencig results.
In our study, PCR assay was helpful for an early diagnosis and guiding the treating clinicians for statrting an accurate therapy. We did nucleotide sequencing to validate our PCR findings. But, we will not recommend nucleotide sequencing for each sample for further confirmation of type of fungus. As per the amplicon size after PCR, they can finalyze whetrher the fungus behind the infection is a mucoral group of fungus or a non-mucoral group of fungus and can start treatment accordingly as the choice of drug is different for both.
Reviewer 2 Report
In my opinión the manuscript is interesting, the introduction is clear, well done and introduces the topic with sufficient background. I think the flow-chart establishes the structure of the study and allows a good understanding of the inclusion criteria of the patients.
I think the text should be reviewed and ordered well, fundamentally the results and the discussion.
Some data to correct:
- In line 132 i think that the word caverrnous is wrong. It should be put cavernous.
- In line 157 donde figura 37 0C and 29 0C, the 0 must be in superscript.
- In line 178 Cytomegalo virus, it should be put cytomegalovirus all together.
- I don’t understand what is “presentation”. In line 207 authors says prior to presentation but really presentation is not defined in material and methods.
- In line 214 where it says 17 (66.7%) it should be put 17 (51.5%). All the numbers should be checked well.
- In line 217 where it says four patients, it should be put also %.
- The arrows in figure 2, do not look good, they should be better highlighted.
- In the table 2, where you put PCR Asaay, it should be put Assay.
- In the table 2, numbers must be revised. In the first row, where you put 16 (56.7%) you should put 16 (53.3%), and where you put 8 (23.3%) you should to put 8 (26.7%). The numbers is necessary to review all of them.
- Table 2 must be completely revised and the parentheses must be placed correctly.
- In line 267 where it says 23 of 66 (39.4%) it should put (34.9%)
- In line 260 where you put couldnot, I should be separated (could not).
- The figure 5 should be improved. The legend where the AUC appears, does not look good.
- The references in the text must be homogenized. In many parts of the text the authors put .[……………], and I think that the punctuation sign must be after closing the bracket [………………..].
- In line 376 appointments 18-21 appear but 16 and 17 do not appear previously.
- In line 390 where you put mucorlaes it must be changed for Mucorales.
- All the references must be reviewed in the text. I think that is better to put the text, behind the references in parentheses or barckets, and then the punctuation mark that is considered (,;.)
Thank you very much
Author Response
Dear Sir
I am sending you the revised manuscript as an attachment also for your reference
Response to Reviewer 2 Comments
Some data to correct:
Point 1:
- In line 132 i think that the word caverrnous is wrong. It should be put cavernous.
Response: As per your suggestion, the term “cavernous” has been changed with “cavernous”. It is shown in track changes.
Pont 2:
- In line 157 donde figura 37 0C and 29 0C, the 0 must be in superscript.
Response: Changes made in the revised manuscript in the track change.
Pont 3:
- In line 178 Cytomegalo virus, it should be put cytomegalovirus all together.
Response: Changes made in the revised manuscript in the track change.
Pont 4:
- I don’t understand what is “presentation”. In line 207 authors says prior to presentation but really presentation is not defined in material and methods.
Response: Sorry for the confusion: Line 206 and 207 in the manuscript states: “Sixteen out of 33 ( 48.5 %) patients were admiited to hospital for COVID-19 treatment prior to presentation for ROCM” . This statement has been changed to : “Sixteen out of 33 ( 48.5%) patients were admitted to hospital for COVID-19 treatment before they attended our institute for ophthalmological evaluation with clinical features suggestive of ROCM” (Line: 213-215)
Pont 5:
- In line 214 where it says 17 (66.7%) it should be put 17 (51.5%). All the numbers should be checked well.
Response: Thanks a lot for correcting the mistake. Changes made in the revised manuscript in the track change mode.
Pont 6:
- In line 217 where it says four patients, it should be put also %.
Response: Changes made in the revised manuscript. 12.1% is now included in the revised manuscript after 4 patients.
Pont 7:
- The arrows in figure 2, do not look good, they should be better highlighted.
Response: Thanks for the remark: Arrows in figure 2 has been changed to red color and made bold to better highlight the findings. The entire figure is now modified as per the suggestion in the revised manuscript.
Pont 8:
- In the table 2, where you put PCR Asaay, it should be put Assay.
Response: Changes made in the revised manuscript in track change mode.
Pont 9:
- In the table 2, numbers must be revised. In the first row, where you put 16 (56.7%) you should put 16 (53.3%), and where you put 8 (23.3%) you should to put 8 (26.7%). The numbers is necessary to review all of them.
Response: Changes incorporated in the revised manuscript. The above corrected values are now incorporated in the revised manuscript. Changes made in table 2 : 53.3% instead of 56.7% and 26.7% instead of 23.3%. All numbers rechecked and found to be correct after rechecking.
Pont 10:
- Table 2 must be completely revised and the parentheses must be placed correctly.
Response: Thanks for the remark: Table 2 has been revised and all changes done. Now parentheses are placed correctly.
Pont 11:
- In line 267 where it says 23 of 66 (39.4%) it should put (34.9%)
Response: Changes are incorporated in the revised manuscript.
Pont 12:
- In line 260 where you put couldnot, I should be separated (could not).
Response: Changes incorporated in the revised manuscript.
Pont 13:
- The figure 5 should be improved. The legend where the AUC appears, does not look good.
Response: Thanks a lot for your valuable feedback. Now we have changed the earlier Figure 5 with the revised one.
Pont 14:
- The references in the text must be homogenized. In many parts of the text the authors put .[……………], and I think that the punctuation sign must be after closing the bracket [………………..].
Response: Changes incorporated in the revised manuscript. Now punctuation sign i.e., full stop is kept after the [………………..] mark in the revised manuscript.
Pont 15:
- In line 376 appointments 18-21 appear but 16 and 17 do not appear previously.
Response: Changes incorporated in the revised manuscript. Thanks a lot for correcting the mistake. Now the order of the references is changed as per your suggestion. Ref 16 and 17 are placed before 18 and accordingly changes are also made in the reference section of the manuscript.
Pont 16:
- In line 390 where you put mucorlaes it must be changed for Mucorales.
Response: Changes incorporated in the revised manuscript.
Pont 17:
- All the references must be reviewed in the text. I think that is better to put the text, behind the references in parentheses or barckets, and then the punctuation mark that is considered (,;.)
Response: Changes incorporated in the revised manuscript as per suggestion. Now, punctuation mark is placed after the square bracket in all parts of manuscript.
Round 2
Reviewer 1 Report
Dear Authors,
Thank you for modifying the manuscript according to the Editor´s suggestions. I think the manucript was improved for its publication.